# Differences in Geochemical Signatures and Petrogenesis between the Van Canh and Ben Giang-Que Son Granitic Rocks in the Southern Kontum Massif, Vietnam

Etsuo Uchida *, Ko Yonezu, Takumi Yokokura and Nasuka Mori

Department of Resources and Environmental Engineering, School of Creative Science and Engineering, Waseda University, Shinjuku-ku, Tokyo 169-8555, Japan; azuenyan@asagi.waseda.jp (K.Y.); tyokokura@toki.waseda.jp (T.Y.); jikishinn@fuji.waseda.jp (N.M.)
* Correspondence: weuchida@waseda.jp

**Abstract:** Permian Ben Giang-Que Son and Triassic Van Canh granitic rocks are widely distributed across the southern Kontum Massif, the basement of which consists mainly of metasedimentary rocks. The Ben Giang-Que Son granitic rocks are classified as I- to S-type and ilmenite-series granitic rocks, while the Van Canh granitic rocks are classified as I-type and magnetite-series granitic rocks. Both granitic rock suites exhibit more or less adakitic properties, suggesting that the subduction of the high-temperature Song Ma Ocean crust, part of the Paleo-Tethys Ocean, beneath the Indochina Block produced adakitic magma. It is hypothesized that the differences between the two granitic rock suites were caused by differences in the quantities of incorporated continental crustal materials and carbon or graphite in clastic sedimentary rocks when their adakitic magma intruded into the continental crust. Based on their high initial Sr isotope ratios, the Ben Giang-Que Son granitic rocks evidently incorporated a higher quantity of continental crustal materials compared to the Van Canh granitic rocks, resulting in the former showing the signatures of ilmenite-series and I- to S-type granitic rocks. Consequently, the Ben Giang-Que Son granitic rocks have relatively high A/CNK ratios and high total Al contents in their biotite, whereas the Van Canh granitic rocks have low A/CNK ratios and low total Al contents in their biotite. The intrusion of the Ben Giang-Que Son granitic rocks caused high-temperature metamorphism, which decomposed some of the carbon or graphite in the surrounding continental crustal materials, such as clastic sedimentary rocks. Meanwhile, the Van Canh granitic rocks, which intruded later than the Ben Giang-Que Son granitic rocks, incorporated smaller quantities of carbon or graphite in continental crustal materials, resulting in them retaining the chemical characteristics of adakitic, magnetite-series, and I-type granitic rocks, different from the Ben Giang-Que Son granitic rocks.

**Keywords:** Van Canh granitic rocks; Ben Giang-Que Son granitic rocks; Kontum Massif; geochemical signatures; magnetic susceptibility; petrogenesis; Vietnam

## 1. Introduction

The tectonic history of mainland Indochina can be explained by the subduction of the Paleo-Tethys Ocean and the amalgamation of the South China, Indochina, and Sibumasu blocks, which separated from the supercontinent of Gondwana [1–18].

The Song Ma Suture, a partial relict of the Paleo-Tethys Ocean that once existed between the South China and Indochina blocks, runs from the northwest to the southeast of northern Vietnam (Figure 1) [5,15,19–21]. The Truong Son Fold Belt is located to the southwest of the Song Ma Suture and was generated by the amalgamation of the South China and Indochina blocks when the Song Ma Ocean was subducted beneath the Indochina Block. The Truong Son Fold Belt continues to the Kontum Massif, which is situated in central Vietnam (Figure 1). Moreover, in the south of the Kontum Massif, there are many Cretaceous

granitic rocks that were formed during the subduction of the Paleo-Pacific Plate, which form the Dalat–Kratie Zone [22].

The Kontum Massif comprises part of the eastern edge of the Indochina Block (Figure 1). Permian Ben Giang-Que Son granitic rocks (ca. 280–260 Ma) and Triassic Van Canh granitic rocks (ca. 251–229 Ma) are widely distributed across the southern part of the Kontum Massif [23–29]. Based on the formation ages of these granitic rocks, most of the Kontum Massif is considered to belong to the Truong Son Fold Belt.

The basic classification of granitic rocks was proposed by Chapell and White [30] and Ishihara [31]. Chappell and White [30] classified granitic rocks as I-type (igneous rock) and S-type (sedimentary rock), based on differences in the source materials. Ishihara [31] classified granitic rocks as magnetite-series and ilmenite-series granite, based on their magnetic susceptibility. Although there are some differences between the two classifications, the former is considered to have been formed under relatively oxidizing conditions, while the latter is considered to have been formed under relatively reducing conditions. The Ben Giang-Que Son granitic rocks are reported to be I-type granitic rocks [32], while the Van Canh granitic rocks are reported to be S-type granitic rocks [29]. However, this study revealed that the Van Canh granitic rocks have high magnetic susceptibility; therefore, they should be classified as magnetite-series and I-type granitic rocks, contradicting their current classification as S-type granitic rocks. In this study, we conducted a detailed investigation of the Van Canh and Ben Giang-Que Son granitic rocks using samples taken from the Kontum and Gia Lai provinces, which occupy the southern part of the Kontum Massif. We performed in situ magnetic susceptibility measurements and collected granitic rock samples for whole-rock chemical composition analysis, Nd–Sr isotope ratio measurements, and biotite chemical composition analysis. On the basis of these data, we aimed to clarify the petrogenesis of both suites of granitic rocks in relation to the tectonic history between the South China Block, the Indochina Block, and the Song Ma Ocean.

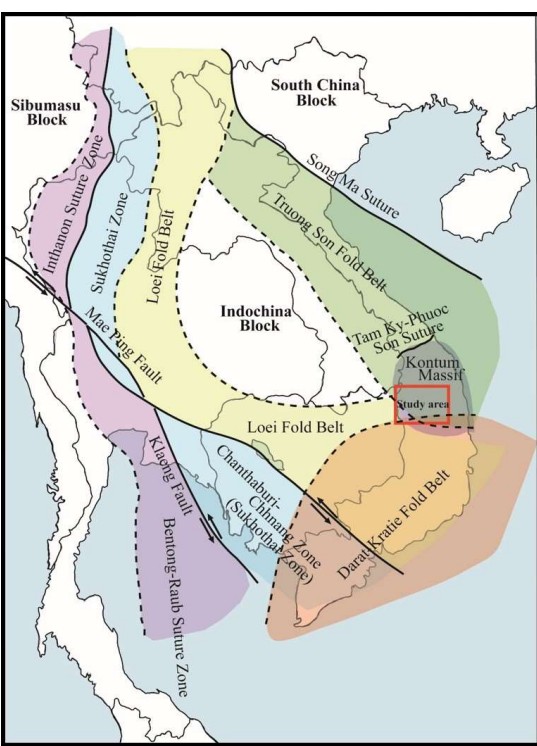

**Figure 1.** Simplified tectonic setting map of southeast Asia. Data obtained from Metcalfe [5], Wang et al. [15], Cheng et al. [19], Kasahara et al. [20], Uchida et al. [21], Hung et al. [28], and this study. The study area is shown by the red square.

## 2. Geological Settings

The northernmost part of Vietnam belongs to the South China Block, while the southern part, which is separated from the northernmost part by the Song Ma Suture, belongs to the Indochina Block. The southern part of Vietnam consists of the Truong Son Belt, the Kontum Massif, and the Dalat Zone (from north to south) (Figure 1) [4,5,13,15,22].

The Kontum Massif is located in central Vietnam and has the largest Precambrian basement in Southeast Asia. The Kontum Massif is delimited on its northern side by the east–west striking Tam Ky-Phuoc Son Suture (Figure 1); however, its other boundaries are not clear.

The basement of the Kontum Massif consists mainly of five different units of metasedimentary rocks, which were deposited in five periods from the Late Paleoproterozoic (1.80–1.65 Ga) to the Late Neoproterozoic–Early Paleozoic eras (0.61–0.51 Ga) [33]. It is considered that most of the Precambrian sediments came from southwestern Laurentia [33].

Precambrian metamorphic rocks are widely distributed across the Kontum Massif, forming the core of the Indochina Block. Ordovician-Silurian and Permian-Triassic high- to ultra-high-temperature metamorphic rocks (amphibolite- to granulite-facies) are also common [23,34–40]. There are four complexes distributed across the Kontum Massif: the Kham Duc, Ngoc Linh, Kan Nak, and Dien Binh complexes (from northwest to southeast) [41]. The Van Canh and Ben Giang-Que Son granitic rocks investigated in this study are mainly distributed in the Kan Nak and Dien Binh complexes. The Kan Nack Complex experienced granulite- to amphibolite-facies metamorphism, while the Ngoc Linh Complex mainly comprises amphibolite-facies metamorphic rocks [33].

## 3. Materials and Methods

The distributions of Van Canh and Ben Giang-Que Son granitic rocks across the Kontum and Gia Lai provinces are shown in Figure 2. Granitic rock samples were collected from roadside outcrops and quarries, and we selected granitic rocks that were as fresh and little weathered as possible. In total, 10 Van Canh and 5 Ben Giang-Que Son granitic rock samples were collected in this study [42]. Table 1 shows the latitudes and longitudes of each sampling site. Magnetic susceptibility was also measured at 10 points across each sampling site using a portable magnetic susceptibility meter (SM30, ZH Instruments, Brno, Czech Republic).

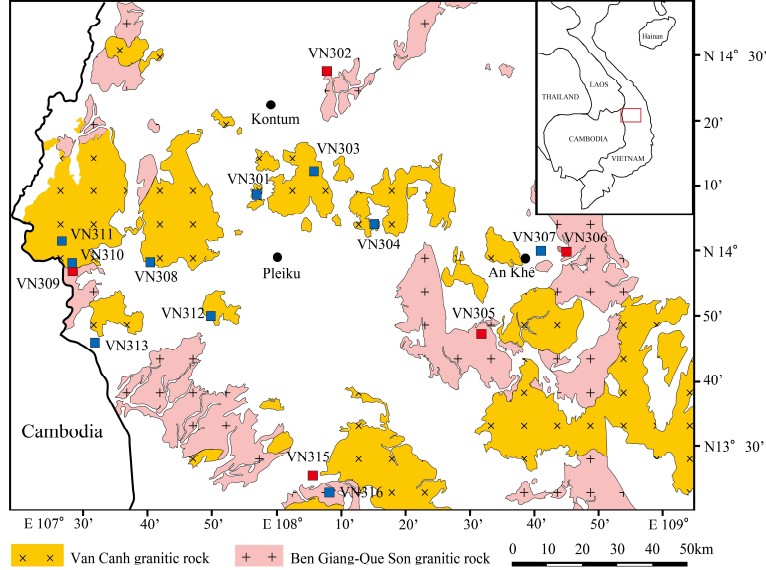

**Figure 2.** Map showing the distributions of Van Canh and Ben Giang-Que Son granitic rocks in the Kontum and Gia Lai provinces, based on the Geological and Mineral Resources Map of Vietnam [42].

**Table 1.** Sampling locations (latitude and longitude) of the Van Canh and Ben Giang-Que Son granitic rock samples.

|  | Sample No. | Latitude | Longitude |
|---|---|---|---|
| Van Canh granitic rock | VN301 | 14°07′52.1″ N | 107°57′36.4″ E |
|  | VN303 | 14°11′10.9″ N | 108°06′16.7″ E |
|  | VN304 | 14°03′11.6″ N | 108°15′19.7″ E |
|  | VN307 | 13°58′45.9″ N | 108°41′14.5″ E |
|  | VN308 | 13°58′11.2″ N | 107°41′05.0″ E |
|  | VN310 | 13°58′24.8″ N | 107°29′09.3″ E |
|  | VN311 | 14°02′06.7″ N | 107°27′21.9″ E |
|  | VN312 | 13°49′57.2″ N | 107°50′22.9″ E |
|  | VN313 | 13°46′18.0″ N | 107°32′32.4″ E |
|  | VN316 | 13°22′47.4″ N | 108°07′50.6″ E |
| Ben Giang-Que Son granitic rock | VN302 | 14°26′09.7″ N | 108°08′16.8″ E |
|  | VN305 | 13°46′01.4″ N | 108°31′26.2″ E |
|  | VN306 | 13°58′12.4″ N | 108°44′31.9″ E |
|  | VN309 | 13°57′31.9″ N | 107°29′06.3″ E |
|  | VN315 | 13°25′37.0″ N | 108°05′36.5″ E |

Thin sections were prepared from the collected granitic rock samples and mineral identification was performed under a polarizing microscope. The granitic rock samples were also ground using a tungsten carbide rod mill (TI-100, Heiko Seisakusho Ltd., Fukushima, Japan) and 5 g of powder from each rock sample was sent to Activation Laboratories Ltd., Ontario, Canada, for whole-rock chemical composition analysis. This analysis was requested according to the "4Litho" litho-geochemistry package. However, the tungsten and cobalt values were excluded from the analytical results owing to contamination during the pulverization of the rock samples using the tungsten carbide rod mill.

The biotite in the granitic rock samples was subjected to chemical composition analysis using an energy dispersive spectrometer (INCA ENERGY, Oxford Instruments, Abingdon, UK), which was attached to a scanning electron microscope (JEOL JSM-6360, Tokyo, Japan) (SEM-EDS) in Waseda University. The thin sections were carbon-coated prior to analysis. The accelerating voltage was 15 kV, and the current was adjusted so that the total X-ray counts on metallic cobalt were 2000 counts/s. The measurement time was around 60 s. The elements measured were Si, Ti, Al, Fe, Mn, Mg, Na, and K and synthetic $SiO_2$, $TiO_2$, $Al_2O_3$, $Fe_2O_3$, $MnO$, $MgO$, natural albite, and K-feldspar were used as the standard materials, respectively. In addition to microscopic observations, the SEM-EDS analysis results were also taken into consideration when determining whether the biotite was chloritized. If the number of K atoms was less than 1.6 on the basis of O = 22, the biotite was considered to be chloritized and was excluded from the results.

Nd–Sr isotope ratio measurements were performed on the collected rock samples at the Research Institute for Humanity and Nature in Kyoto, Japan. The separation of Nd and Sr from the collected rock samples was performed according to the method of Na et al. [43]. A multi-collector inductively coupled plasma mass spectrometer (MC–ICP–MS; NEPTUNE, Thermo Fisher Scientific Inc., Waltham, MA, USA) was used for the analysis. The measured $^{87}Sr/^{86}Sr$ and $^{143}Nd/^{144}Nd$ isotope ratios were corrected using the abundance ratios of $^{86}Sr/^{88}Sr = 0.1194$ [44] and $^{146}Nd/^{144}Nd = 0.7219$ [45] in nature. The $^{87}Sr/^{86}Sr$ ratio for the Sr standard sample (NIST SRM 987), which was measured at the same time as the unknown samples, was $0.710286 \pm 0.000011$ (2σ) (n = 5) and the $^{143}Nd/^{144}Nd$ ratio for the Nd standard sample (JNdi-1) was $0.511934 \pm 0.000013$ (2σ) (n = 6). The Sr and Nd isotope ratios for the measured rock samples were corrected using the isotope ratios of $^{87}Sr/^{86}Sr = 0.710250$ (NIST SRM 987) [46] and $^{143}Nd/^{144}Nd = 0.512115$ (JNdi-1) [47], respectively.

## 4. Results

### 4.1. Petrographic Description and Constituent Minerals

Photographs and photomicrographs under a polarizing microscope of representative Van Canh and Ben Giang-Que Son granitic rock samples are shown in Figure 3. The rock sampling locations (latitude and longitude) and constituent minerals are summarized in Tables 1 and 2, respectively.

### Van Canh granitic rock

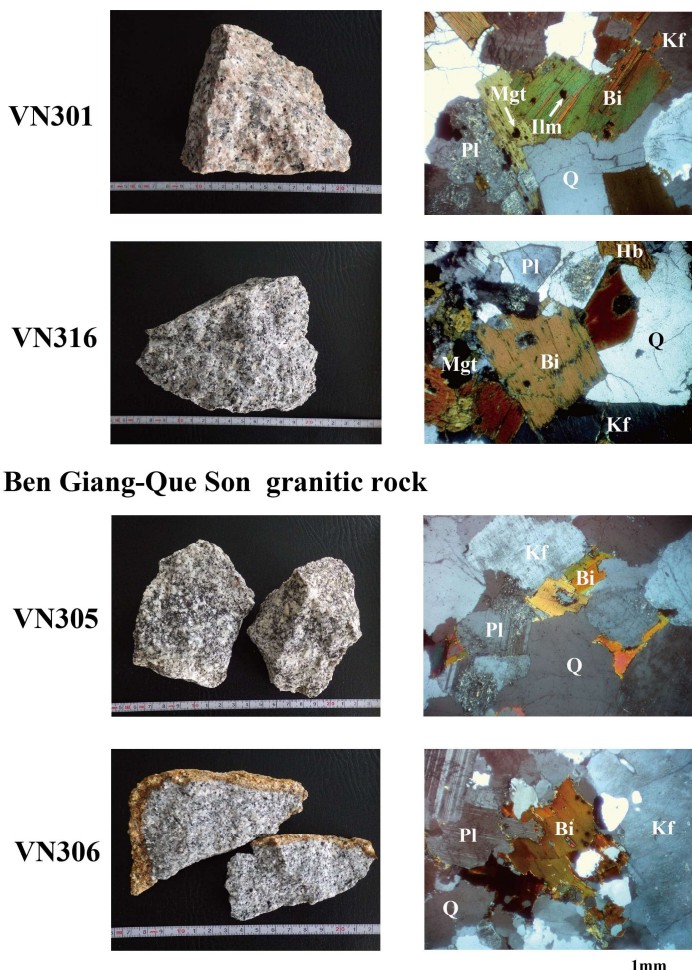

### Ben Giang-Que Son granitic rock

**Figure 3.** Photographs of representative samples of the Van Canh and Ben Giang-Que Son granitic rocks (**left**) and photomicrographs of the corresponding thin sections under cross-polarized light (**right**). Abbreviations: Q, quartz; Pl, plagioclase; Kf, K-feldspar; Bi, biotite; Hb, hornblende; Mgt, magnetite; Ilm, ilmenite.

**Table 2.** Constituent minerals of the Van Canh and Ben Giang-Que Son granitic rock samples.

| Granitic Body | Sample No. | Rock Type | Q | Pl | Kf | Bi | Hb | Zr | Ap | Mu | Ti | Op | Ep | Ru | Cpx | Tour | Alla | Cal | Remarks |
|---|---|---|---|---|---|---|---|---|---|---|---|---|---|---|---|---|---|---|---|
| Van Canh granitic rock | VN301 | Biotite Granite | ○ | ○ | ◎ | ○ | | – | – | | – | – | | | | | | | Bi is partly altered. |
| | VN303 | Biotite Granite | ◎ | ◎ | ◎ | △ | | | – | – | – | – | | – | | | | | Pl and Bi are altered. |
| | VN304 | Hornblend Biotite Granite | ○ | ○ | ◎ | △ | – | – | | – | – | | | | | | | | |

**Table 2.** *Cont.*

| Granitic Body | Sample No. | Rock Type | Q | Pl | Kf | Bi | Hb | Zr | Ap | Mu | Ti | Op | Ep | Ru | Cpx | Tour | Alla | Cal | Remarks |
|---|---|---|---|---|---|---|---|---|---|---|---|---|---|---|---|---|---|---|---|
| Van Canh granitic rock | VN307 | Hornblend Biotite Granite | ○ | ◎ | ○ | ○ | – | △ | – | | | – | – | | | – | – | | |
| | VN308 | Hornblend Biotite Granite | ○ | ○ | ◎ | △ | – | – | – | | | – | | | | | | | Bi is altered. |
| | VN310 | Biotite Granite | ◎ | ○ | ◎ | △ | | – | | | – | – | | | | | | | Bi is almost altered. |
| | VN311 | Hornblend Biotite Granite | ○ | ○ | ◎ | △ | – | – | | | – | – | | | | | | | Bl is partly altered. |
| | VN312 | Biotite Granite | ○ | ○ | ◎ | △ | | – | – | | – | – | | | | | | | Bl is partly altered. |
| | VN313 | Diorite | ○ | ◎ | ○ | ○ | ○ | – | – | – | – | – | – | | – | | | | Bl is partly altered. |
| | VN316 | Biotite Granite | ○ | ○ | ○ | △ | ○ | – | – | | – | ○ | | | | | | | |
| Ben Giang-Que Son granitic rock | VN302 | Biotite Granite | ○ | ◎ | ○ | ○ | | – | – | △ | | – | | | | | | | Bi is almost altered. |
| | VN305 | Biotite Granite | ◎ | ○ | ◎ | ○ | | – | – | | | – | | | | | | | |
| | VN306 | Biotite Granite | ◎ | ○ | ◎ | ○ | | – | – | | | – | | | | | | | Pl and Bi are altered. |
| | VN309 | Biotite Granite | ○ | ◎ | ◎ | ○ | | – | – | | | – | | | | | | | Bl is partly altered. |
| | VN315 | Biotite Granite | ◎ | ◎ | ◎ | △ | | – | – | – | | – | | | | | | | Bi is almost altered. |

Modal proportions: ◎, >30 vol%; ○, 30–10 vol%; △, 10–2 vol%; –, <2 vol%. Abbreviations: Q, quartz; Pl, plagioclase; Kf, K-feldspar; Bi, biotite; Hb, hornblende; Ch, chlorite; Zr, zircon; Ap, apatite; Mu, muscovite; Ti, titanite; Op, opaque minerals; Ep, epidote; Ru, rutile; Cpx, clinopyroxene; Tour, tourmaline; Alla, allanite; Cal, calcite.

The Van Canh granitic rocks were coarse- to medium-grained and gray-white to slightly pink in color. The major constituent minerals were K-feldspar, plagioclase, quartz, biotite, and amphibole. Zircon, apatite, titanite, opaque minerals (magnetite and ilmenite), and epidote were also observed as minor constituent minerals. The chloritization of biotite was recognized in some samples.

The Ben Giang-Que Son granitic rocks were also medium-grained and gray-white to slightly pink in color. The main constituent minerals were K-feldspar, plagioclase, quartz, and biotite. Zircon, apatite, opaque minerals (ilmenite and pyrite), and muscovite were also observed as minor constituent minerals. The chloritization of biotite was recognized in some samples.

The Van Canh granitic rocks contained hornblende and small amounts of titanite, whereas the Ben Giang-Que Son granitic rocks did not contain these minerals but were accompanied by small amounts of muscovite. These facts indicated that the Van Canh granitic rocks were I-type and magnetite-series granitic rock, while the Ben Giang-Que Son granitic rocks were I- to S-type and ilmenite-series granitic rock [48,49].

### 4.2. Magnetic Susceptibility

The classification of magnetite-series and ilmenite-series granitic rocks was first proposed by Ishihara [31].

The magnetic susceptibility of samples from both granitic rock suites was measured at each outcrop, as shown in Figure 4. Because the average magnetic susceptibility of the Van Canh granitic rocks was higher than $3 \times 10^{-3}$ SI units, they were classified as magnetite-series granitic rocks [48]. However, sample VN307 showed magnetic susceptibility values lower than $3 \times 10^{-3}$ SI units at several points, which could have been due to the weathering of the granitic rocks on the ground surface. In contrast, all of the Ben Giang-Que Son granitic rocks showed magnetic susceptibility values lower than $3 \times 10^{-3}$ SI units and were classified as ilmenite-series granitic rocks [48].

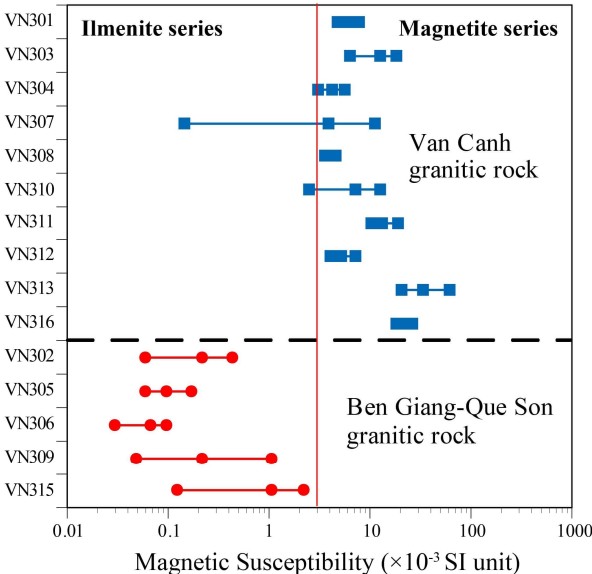

**Figure 4.** Magnetic susceptibility of the Van Canh and Ben Giang-Que Son granitic rocks from each sampling site. The minimum (**left**), mean (**center**), and maximum (**right**) values are shown.

### 4.3. Whole-Rock Chemical Composition

The whole-rock chemical composition analysis results for the Van Canh and Ben Giang-Que Son granitic rock samples are summarized in Table 3.

**Table 3.** Whole-rock chemical compositions of the Van Canh and Ben Giang-Que Son granitic rock samples.

| Locality | | Van Canh Granitic Rock | | | | | | | | | | Ben Giang-Que Son Granitic Rock | | | | |
|---|---|---|---|---|---|---|---|---|---|---|---|---|---|---|---|---|
| Sample No. | | VN301 | VN303 | VN304 | VN307 | VN308 | VN310 | VN311 | VN312 | VN313 | VN316 | VN302 | VN305 | VN306 | VN309 | VN315 |
| SiO$_2$ | % | 71.46 | 66.92 | 73.45 | 68.22 | 72.45 | 72.34 | 72.34 | 72.76 | 60.86 | 63.65 | 72.87 | 70.14 | 72.56 | 68.26 | 73.19 |
| Al$_2$O$_3$ | % | 13.43 | 15.22 | 13.55 | 14.64 | 13.14 | 13.44 | 14.58 | 13.21 | 15.54 | 16.03 | 14 | 14.97 | 13.55 | 14.36 | 13.45 |
| Fe$_2$O$_3$(T) | % | 2.46 | 3.22 | 1.84 | 3.72 | 2.09 | 2.07 | 1.46 | 2.06 | 5.54 | 4.89 | 1.37 | 2.7 | 2.23 | 4.68 | 1.94 |
| MnO | % | 0.065 | 0.047 | 0.049 | 0.053 | 0.075 | 0.032 | 0.055 | 0.062 | 0.082 | 0.074 | 0.023 | 0.037 | 0.027 | 0.05 | 0.061 |
| MgO | % | 0.49 | 0.87 | 0.22 | 0.81 | 0.12 | 0.41 | 0.45 | 0.47 | 3.04 | 1.81 | 0.32 | 0.73 | 0.3 | 1.67 | 0.28 |
| CaO | % | 1.66 | 3.19 | 1.16 | 3.26 | 0.65 | 1.72 | 1.99 | 1.66 | 4.94 | 4.38 | 1.37 | 2.43 | 1.09 | 3.01 | 0.58 |
| Na$_2$O | % | 3.02 | 3.59 | 3.36 | 3.08 | 3.43 | 2.58 | 3.34 | 3.18 | 2.87 | 3.19 | 3.15 | 2.81 | 2.24 | 3.38 | 3.74 |
| K$_2$O | % | 4.72 | 4.11 | 5.1 | 3.18 | 5.71 | 5.01 | 4.43 | 4.11 | 2.49 | 3.68 | 4.72 | 4.78 | 5.95 | 1.65 | 4.82 |
| TiO$_2$ | % | 0.368 | 0.43 | 0.184 | 0.504 | 0.228 | 0.227 | 0.212 | 0.267 | 0.807 | 0.838 | 0.24 | 0.412 | 0.288 | 0.525 | 0.18 |
| P$_2$O$_5$ | % | 0.08 | 0.1 | 0.02 | 0.12 | 0.01 | 0.06 | 0.04 | 0.05 | 0.23 | 0.17 | 0.06 | 0.13 | 0.09 | 0.05 | 0.01 |
| LOI | % | 1.07 | 1.66 | 0.61 | 1.19 | 0.9 | 0.74 | 1.08 | 1 | 1.94 | 1.28 | 1.26 | 0.75 | 1.18 | 1.34 | 1.24 |
| Total | % | 98.83 | 99.36 | 99.55 | 98.77 | 98.8 | 98.63 | 98.99 | 98.82 | 98.34 | 100 | 99.39 | 99.88 | 99.5 | 98.98 | 99.5 |
| Sc | ppm | 6 | 6 | 7 | 7 | 10 | 3 | 4 | 6 | 13 | 11 | 3 | 5 | 6 | 9 | 6 |
| Be | ppm | 3 | 2 | 2 | 2 | 3 | 2 | 2 | 4 | 2 | 2 | 3 | 3 | 2 | 3 | 3 |
| V | ppm | 20 | 51 | 11 | 40 | <5 | 20 | 18 | 22 | 110 | 88 | 14 | 26 | 9 | 61 | 6 |
| Ba | ppm | 378 | 714 | 317 | 694 | 93 | 491 | 964 | 334 | 840 | 961 | 860 | 624 | 423 | 86 | 690 |
| Sr | ppm | 159 | 326 | 95 | 278 | 15 | 244 | 245 | 125 | 785 | 424 | 243 | 297 | 129 | 191 | 97 |
| Y | ppm | 26 | 20 | 24 | 15 | 36 | 12 | 10 | 39 | 17 | 35 | 11 | 11 | 18 | 8 | 43 |
| Zr | ppm | 191 | 166 | 173 | 211 | 335 | 136 | 94 | 158 | 192 | 327 | 149 | 224 | 221 | 237 | 275 |
| Cr | ppm | <20 | <20 | <20 | <20 | <20 | <20 | <20 | <20 | 60 | 20 | <20 | <20 | <20 | 40 | <20 |
| Ni | ppm | <20 | <20 | <20 | <20 | <20 | <20 | <20 | <20 | 20 | <20 | <20 | <20 | <20 | 20 | <20 |
| Cu | ppm | <10 | <10 | <10 | <10 | <10 | <10 | <10 | <10 | < 10 | 10 | <10 | <10 | <10 | 30 | <10 |
| Zn | ppm | 40 | 30 | 50 | <30 | 50 | <30 | <30 | <30 | 60 | 50 | <30 | 40 | 50 | 60 | 40 |
| Ga | ppm | 16 | 21 | 17 | 18 | 17 | 15 | 13 | 16 | 19 | 18 | 19 | 19 | 18 | 19 | 18 |
| Ge | ppm | 1 | 2 | 2 | 1 | 2 | 1 | 1 | 2 | 1 | 1 | 1 | 2 | 2 | 1 | 2 |
| As | ppm | <5 | <5 | < 5 | <5 | <5 | <5 | <5 | <5 | <5 | <5 | <5 | <5 | <5 | <5 | <5 |
| Rb | ppm | 220 | 179 | 195 | 109 | 194 | 188 | 109 | 248 | 78 | 137 | 247 | 185 | 270 | 141 | 210 |
| Nb | ppm | 11 | 14 | 8 | 7 | 13 | 5 | 4 | 13 | 9 | 15 | 8 | 8 | 13 | 16 | 16 |
| Mo | ppm | <2 | <2 | <2 | <2 | <2 | <2 | <2 | <2 | <2 | <2 | <2 | <2 | <2 | <2 | <2 |
| Ag | ppm | <0.5 | <0.5 | <0.5 | <0.5 | <0.5 | <0.5 | <0.5 | <0.5 | <0.5 | <0.5 | <0.5 | <0.5 | <0.5 | <0.5 | <0.5 |
| In | ppm | <0.2 | <0.2 | <0.2 | <0.2 | <0.2 | <0.2 | <0.2 | <0.2 | <0.2 | <0.2 | <0.2 | <0.2 | <0.2 | <0.2 | <0.2 |
| Sn | ppm | 3 | 2 | 2 | 1 | 2 | 1 | 1 | 3 | 1 | 2 | 5 | 3 | 4 | 3 | 3 |

**Table 3.** *Cont.*

| Locality | | Van Canh Granitic Rock | | | | | | | | | | Ben Giang-Que Son Granitic Rock | | | | |
|---|---|---|---|---|---|---|---|---|---|---|---|---|---|---|---|---|
| Sample No. | | VN301 | VN303 | VN304 | VN307 | VN308 | VN310 | VN311 | VN312 | VN313 | VN316 | VN302 | VN305 | VN306 | VN309 | VN315 |
| Sb | ppm | <0.5 | <0.5 | <0.5 | <0.5 | <0.5 | <0.5 | <0.5 | <0.5 | <0.5 | <0.5 | <0.5 | <0.5 | <0.5 | <0.5 | <0.5 |
| Cs | ppm | 3.7 | 1.8 | 2.4 | 2.2 | 2.8 | 3.5 | 1.3 | 7.2 | 1.7 | 2.4 | 8.4 | 5.7 | 3.8 | 6.9 | 2.4 |
| La | ppm | 48.8 | 38.5 | 62.8 | 44.4 | 134 | 47.8 | 19.4 | 42.3 | 39.7 | 60.1 | 46 | 75.2 | 122 | 33 | 61.5 |
| Ce | ppm | 99.5 | 75.3 | 126 | 85.5 | 250 | 89.7 | 32.9 | 83.4 | 78.7 | 118 | 81.9 | 147 | 223 | 63.6 | 128 |
| Pr | ppm | 11.3 | 8.35 | 14.1 | 9.22 | 26.6 | 9.31 | 3.35 | 8.99 | 9.13 | 13.2 | 8.29 | 16.5 | 28.8 | 6.81 | 14.6 |
| Nd | ppm | 40.2 | 30.1 | 49.8 | 32.3 | 93.5 | 30.9 | 11.4 | 32.3 | 34.8 | 49.7 | 27.4 | 58.8 | 101 | 24.1 | 53 |
| Sm | ppm | 7.2 | 5.8 | 9.2 | 5.4 | 14.2 | 5.1 | 2.1 | 6.2 | 6.2 | 9.3 | 4.4 | 9.2 | 17.9 | 4.6 | 10.4 |
| Eu | ppm | 0.98 | 1.1 | 0.47 | 1.12 | 0.95 | 0.73 | 0.83 | 0.7 | 1.39 | 1.39 | 0.59 | 1.02 | 1.18 | 0.66 | 1.11 |
| Gd | ppm | 5.5 | 4.5 | 6.9 | 4 | 10 | 3.3 | 1.8 | 5.3 | 4.6 | 7.3 | 3 | 5.3 | 11.2 | 3.4 | 8.2 |
| Tb | ppm | 0.8 | 0.7 | 0.9 | 0.5 | 1.3 | 0.4 | 0.3 | 0.9 | 0.6 | 1.1 | 0.4 | 0.6 | 1.2 | 0.4 | 1.3 |
| Dy | ppm | 4.4 | 3.8 | 5 | 2.8 | 7 | 2.2 | 1.6 | 5.4 | 3.1 | 6.3 | 2.1 | 2.6 | 4.9 | 1.9 | 7.7 |
| Ho | ppm | 0.9 | 0.7 | 0.9 | 0.5 | 1.3 | 0.4 | 0.3 | 1.2 | 0.6 | 1.2 | 0.4 | 0.4 | 0.7 | 0.3 | 1.5 |
| Er | ppm | 2.6 | 2 | 2.5 | 1.4 | 3.6 | 1.1 | 0.9 | 3.8 | 1.6 | 3.4 | 1 | 1.1 | 1.6 | 0.8 | 4.3 |
| Tm | ppm | 0.38 | 0.3 | 0.35 | 0.19 | 0.51 | 0.16 | 0.14 | 0.63 | 0.24 | 0.49 | 0.15 | 0.14 | 0.2 | 0.12 | 0.66 |
| Yb | ppm | 2.3 | 1.9 | 2.2 | 1.1 | 3.3 | 1.1 | 0.9 | 4.3 | 1.5 | 3.1 | 0.9 | 0.8 | 1.1 | 0.8 | 4.8 |
| Lu | ppm | 0.34 | 0.29 | 0.34 | 0.16 | 0.5 | 0.16 | 0.14 | 0.7 | 0.22 | 0.47 | 0.14 | 0.11 | 0.17 | 0.12 | 0.78 |
| Hf | ppm | 4.9 | 4.4 | 4.7 | 4.8 | 7.7 | 3.3 | 2.2 | 4.3 | 4.4 | 7.5 | 3.6 | 5.4 | 5.6 | 5.8 | 8.5 |
| Ta | ppm | 1.6 | 1.8 | 1 | 0.8 | 1.4 | 1 | 0.8 | 2.1 | 0.8 | 1.3 | 1.7 | 1 | 1.3 | 1.9 | 3 |
| Tl | ppm | 1.1 | 0.9 | 1 | 0.6 | 0.9 | 0.9 | 0.5 | 1.1 | 0.5 | 0.6 | 1.4 | 1 | 1.5 | 0.9 | 1 |
| Pb | ppm | 20 | 24 | 22 | 14 | 27 | 25 | 16 | 21 | 17 | 15 | 45 | 25 | 42 | 16 | 24 |
| Bi | ppm | <0.4 | <0.4 | <0.4 | <0.4 | <0.4 | <0.4 | <0.4 | <0.4 | <0.4 | <0.4 | 0.6 | 1 | <0.4 | <0.4 | <0.4 |
| Th | ppm | 28.6 | 13.8 | 25.2 | 12.7 | 26 | 37.7 | 11.5 | 33.8 | 9.7 | 18.7 | 29.9 | 28.3 | 64.6 | 16.6 | 29.3 |
| U | ppm | 4.7 | 1.8 | 3.7 | 1.9 | 3.9 | 10.2 | 1.3 | 10.2 | 1.7 | 2.7 | 7.9 | 3.7 | 4.7 | 4.6 | 4.9 |

A total alkali versus $SiO_2$ (TAS) diagram for the granitic rock samples, based on the analytical results, is shown in Figure 5 [50,51]. On the basis of these results, many of the Van Canh granitic rocks were classified as granite, but there were some rocks showing diorite (VN313) to granodiorite (VN303, VN307, and VN316) compositions. In contrast, most of the Ben Giang-Que Son granitic rocks were classified as granite and only one sample (VN309) was classified as granodiorite.

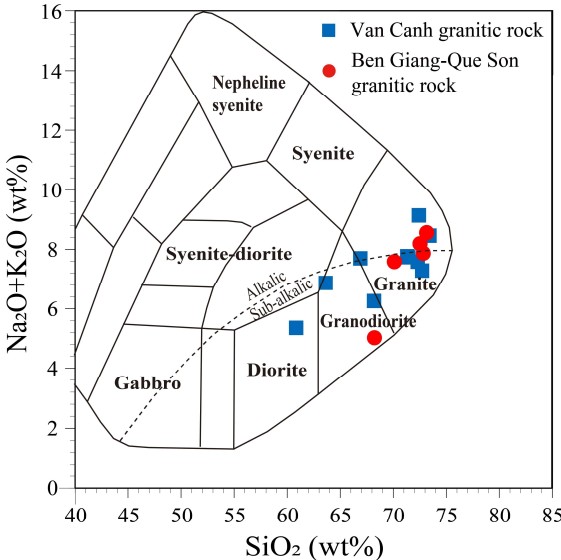

**Figure 5.** Classification of the Van Canh and Ben Giang-Que Son granitic rock samples, on the basis of a total alkali versus $SiO_2$ (TAS) diagram. The classification boundaries are from Cox et al. [50] and Wilson [51]. The dashed line is a boundary between alkalic and non-alkalic rocks.

On the $Al_2O_3/(Na_2O + K_2O)$ versus $Al_2O_3/(CaO + Na_2O + K_2O)$ (A/NK versus A/CNK) diagram [52] (Figure 6), the Van Canh granitic rocks were plotted in the metaluminous to peraluminous region with $Al_2O_3/(CaO + Na_2O + K_2O)$ molar ratios <1.1, so they were classified as I-type granitic rocks [52]. The Ben Giang-Que Son granitic rocks were all plotted in the peraluminous region and were classified as I-type to S-type granitic rocks, with $Al_2O_3/(CaO + Na_2O + K_2O)$ molar ratios of around 1.1.

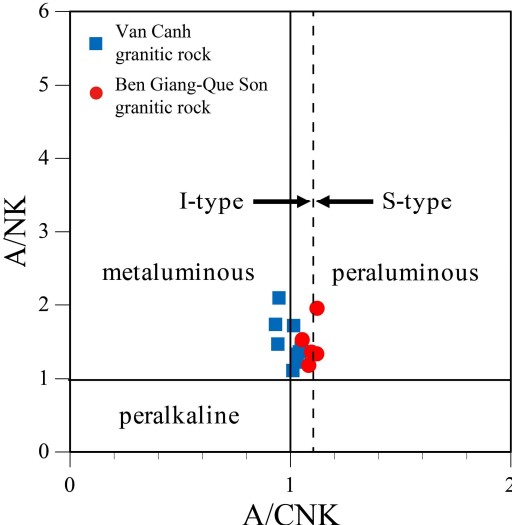

**Figure 6.** Al$_2$O$_3$/(Na$_2$O + K$_2$O) versus Al$_2$O$_3$/(CaO + Na$_2$O + K$_2$O) (A/NK versus A/CNK) diagram for the Van Canh and Ben Giang-Que Son granitic rock samples, showing the classification of I- and S-type granitic rocks and the classification of metaluminous and peraluminous rocks [52]. The dashed line is a boundary between I-type and S-type granitic rocks.

On the Zr versus 10,000 × Ga/Al diagram [53] (Figure 7), two Van Canh and one Ben Giang-Que Son granitic rock samples were plotted in the A-type region, while the others were plotted in the I- and S-type regions. Combined with the data from the Al$_2$O$_3$/(Na$_2$O + K$_2$O) versus Al$_2$O$_3$/(CaO + Na$_2$O + K$_2$O) diagram, the Van Canh samples were classified as I-type granitic rocks, while the Ben Giang-Que Son samples were classified as I- to S-type granitic rocks.

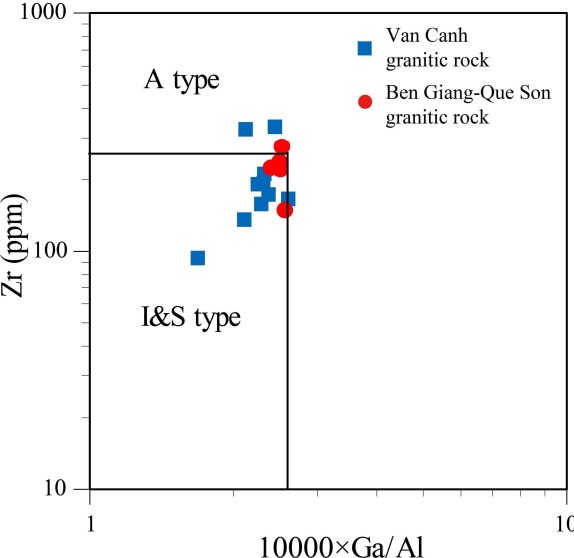

**Figure 7.** Zr versus 10,000 × Ga/Al diagram for the Van Canh and Ben Giang-Que Son granitic rock samples, showing the classification of I-, S-, and A-type granitic rocks [53].

On the basis of the Rb versus (Yb + Ta) tectonic classification diagram of Pearce [54] (Figure 8), the Van Canh granitic rocks were classified as volcanic arc to syn-collisional granitic rocks, while most of the Ben Giang-Que Son granitic rocks were classified as syn-collisional granitic rocks.

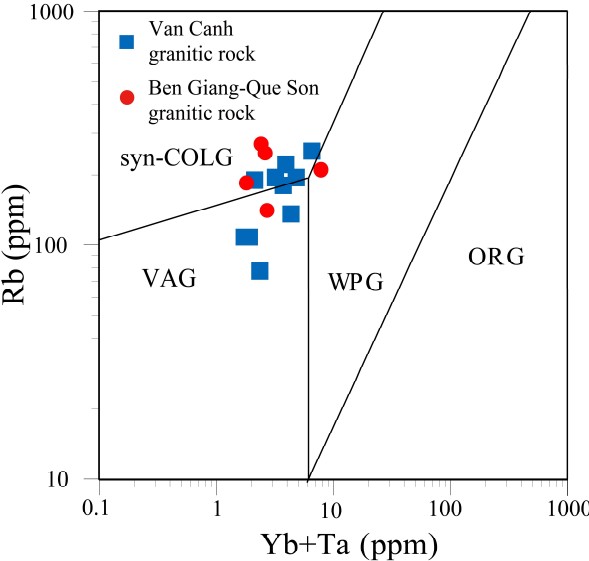

**Figure 8.** Tectonic setting classification diagram for the Van Canh and Ben Giang-Que Son granitic rock samples (from Pearce et al. [54]). Abbreviations: VAG, volcanic arc granite; syn-COLG, syn-collision granite; WPG, within plate granite; ORG, ocean ridge granite.

On the Sr/Y versus Y diagram [55], which distinguishes adakitic and non-adakitic rocks (Figure 9), some of the Van Canh and Ben Giang-Que Son granitic rocks were plotted in the adakitic field. Overall, they also exhibited slightly higher Sr/Y ratios than those found in typical non-adakitic rocks. This indicated the higher or lower degrees of involvement of adakitic magma in both granitic rock suites.

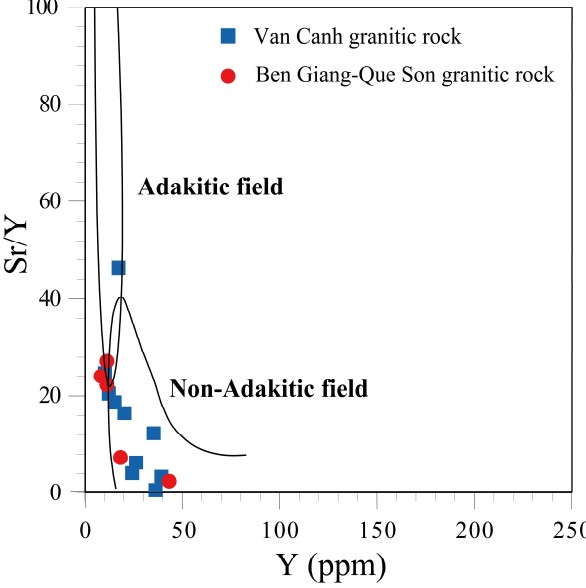

**Figure 9.** Classification of adakitic and non-adakitic rocks in the Van Canh and Ben Giang-Que Son granitic rock samples using the Sr/Y versus Y diagram (Defant and Drummond [55]).

Chondrite-normalized rare earth element (REE) patterns [56] were constructed using the values from MaDonough and Sun [57] (Figure 10). The results revealed that all of the Ben Giang-Que Son granitic rocks exhibited negative Eu anomalies. While many of the Van Canh granitic rock samples also exhibited negative Eu anomalies, these anomalies were smaller in some samples, such as VN307 and VN311, whereas one sample VN313 showed a slightly positive Eu anomaly, which may indicate an accumulation of plagioclase.

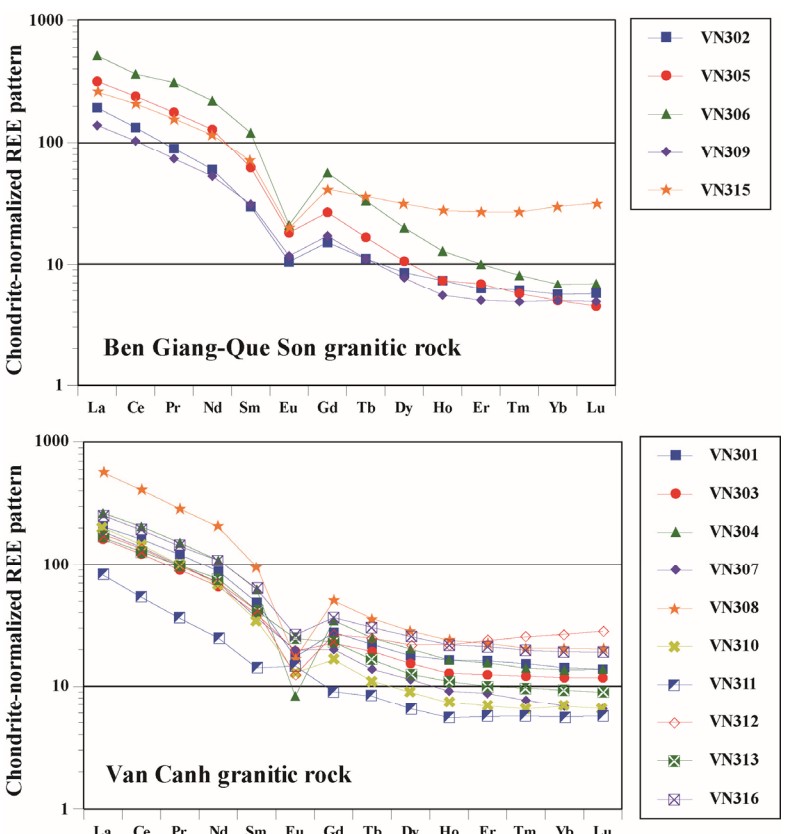

**Figure 10.** Chondrite-normalized rare earth element (REE) patterns [56] of the Van Canh and Ben Giang-Que Son granitic rock samples. The chemical compositions of the chondrite were taken from McDonough and Sun [57].

*4.4. Chemical Compositions of Biotite*

The biotite in the Ben Giang-Que Son granitic rocks, except for one point, exhibited high total Al contents (2.6–3.4; on the basis of O = 22) and low Mg/(Mg + Fe) molar ratios (0.2–0.45) (Figure 11). In contrast, the biotite in the Van Canh granitic rocks tended to have low total Al contents (2.25–2.95; on the basis of O = 22) and high Mg/(Mg + Fe) molar ratios (0.3–0.65).

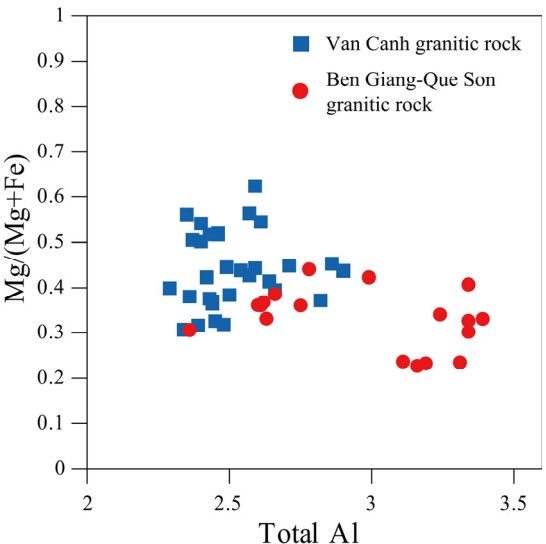

**Figure 11.** Relationship between the Mg/(Mg + Fe) molar ratios and total Al contents in the biotite (based on O = 22) of the Van Canh and Ben Giang-Que Son granitic rock samples.

### 4.5. Nd–Sr Isotope Ratios

The results of the Nd–Sr isotope ratio analysis are summarized in Table 4. In addition, using the zircon U-Pb ages of ca. 240 Ma for the Van Canh granitic rocks [29] and ca. 270 Ma for the Ben Giang-Que Son granitic rocks [28], the calculated initial Nd, $(^{143}Nd/^{144}Nd)_i$, and Sr, $(^{87}Sr/^{86}Sr)_i$ isotope ratios are shown in Figure 12. The $(^{143}Nd/^{144}Nd)_i$ ratios of the Van Canh and Ben Giang-Que Son granitic rocks were 0.511872–0.512216 and 0.511820–0.511984, respectively, showing that there were no significant differences. In contrast, the $(^{87}Sr/^{86}Sr)_i$ ratios of the Van Canh and Ben Giang-Que Son granitic rocks were 0.706821–0.716976 and 0.709143–0.729246, respectively, showing that the Ben Giang-Que Son granitic rocks evidently had higher initial values.

**Table 4.** Results of Sr and Nd isotope ratio measurements for the Van Canh and Ben Giang-Que Son granitic rock samples.

| | Sample No. | U-Pb Age (Ma) | $^{87}Sr/^{86}Sr$ | $\pm 1\sigma$ | $^{87}Rb/^{86}Sr$ | $(^{87}Sr/^{86}Sr)_i$ | $^{143}Nd/^{144}Nd$ | $\pm 1\sigma$ | $^{147}Sm/^{144}Nd$ | $(^{143}Nd/^{144}Nd)_i$ |
|---|---|---|---|---|---|---|---|---|---|---|
| | VN301 | | 0.722798 | 0.000007 | 3.91085 | 0.709447 | 0.512147 | 0.000007 | 0.112944 | 0.511969 |
| | VN303 | | 0.720207 | 0.000009 | 3.92101 | 0.706821 | 0.512063 | 0.000006 | 0.121509 | 0.511872 |
| | VN304 | | 0.729589 | 0.000008 | 36.7688 | 0.709770 | 0.512135 | 0.000005 | 0.116497 | 0.511952 |
| Van Canh | VN307 | | 0.718747 | 0.000007 | 1.25699 | 0.714456 | 0.512057 | 0.000005 | 0.105424 | 0.511892 |
| granitic rock | VN310 | 240 | 0.723084 | 0.000008 | 1.78920 | 0.716976 | 0.512073 | 0.000007 | 0.104079 | 0.511910 |
| | VN311 | | 0.710642 | 0.000004 | 1.10690 | 0.706863 | 0.512398 | 0.000007 | 0.116171 | 0.512216 |
| | VN312 | | 0.729153 | 0.000010 | 5.61124 | 0.709998 | 0.512211 | 0.000007 | 0.121047 | 0.512021 |
| | VN313 | | 0.710342 | 0.000008 | 0.280503 | 0.709385 | 0.512204 | 0.000006 | 0.112351 | 0.512028 |
| | VN316 | | 0.712022 | 0.000009 | 0.912302 | 0.708907 | 0.512244 | 0.000005 | 0.118003 | 0.512059 |
| | VN302 | | 0.721474 | 0.000008 | 2.87263 | 0.710439 | 0.512163 | 0.000005 | 0.101265 | 0.511984 |
| Ben | VN305 | | 0.720205 | 0.000008 | 2.73699 | 0.709691 | 0.512037 | 0.000004 | 0.098663 | 0.511862 |
| Giang-Que | VN306 | 270 | 0.741282 | 0.000012 | 3.13335 | 0.729246 | 0.512017 | 0.000004 | 0.111757 | 0.511820 |
| Son granitic rock | VN309 | | 0.727150 | 0.000009 | 4.19688 | 0.711028 | 0.512073 | 0.000005 | 0.120362 | 0.511860 |
| | VN315 | | 0.732672 | 0.000011 | 6.12512 | 0.709143 | 0.512140 | 0.000007 | 0.123741 | 0.511921 |

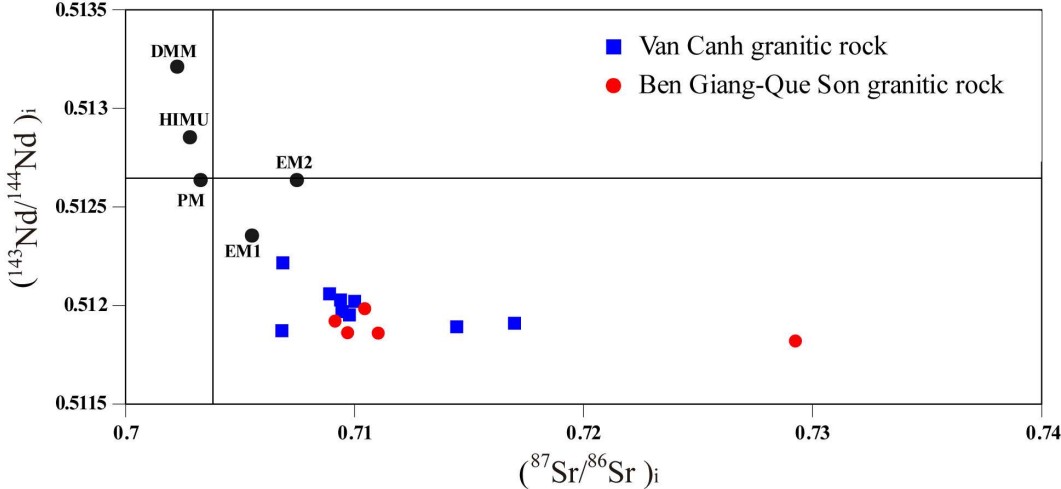

**Figure 12.** $(^{87}Sr/^{86}Sr)_i$ versus $(^{143}Nd/^{144}Nd)_i$ diagram for the Van Canh and Ben Giang-Que Son granitic rock samples, calculated using the zircon U–Pb ages of ca. 240 Ma for the Van Canh granitic rocks [29] and ca. 270 Ma for the Ben Giang-Que Son granitic rocks [28]. The initial Nd and Sr isotope ratio data (black dot) for depleted MORB mantle (DMM), enriched mantle 1 (EM1), enriched mantle 2 (EM2), high μ mantle (HIMU), and primitive mantle (PM) are from Faure and Mensing [46] and Schaefer [58].

### 5. Discussion

The Ben Giang-Que Son granitic rocks had intermediate compositions between I-type and S-type granitic rocks and were classified as ilmenite-series granitic rocks. Their initial Sr isotope ratio values indicated that they incorporated large quantities of continental crustal materials (Figure 12) [46,58]. Continental crustal materials, such as clastic sedimentary rocks, usually contain carbon or graphite [48,49]. Jiang et al. [33] demonstrated that

the Kontum Massif basement consists mainly of different units of metasedimentary rocks, which were derived from clastic sedimentary rocks deposited in five periods. Magma becomes reductive when these continental crustal materials are incorporated. In magma that has become reductive due to the incorporation of carbon or graphite, $Eu^{3+}$ is reduced to $Eu^{2+}$ and is incorporated into plagioclase. As plagioclase is removed from the magma by differentiation, $Eu^{3+}$ in the magma becomes relatively low, resulting in negative Eu anomalies in chondrite-normalized REE patterns. In addition, the incorporation of peraluminous continental crustal materials, such as clastic sedimentary rocks, increases A/CNK ratios and total Al contents in biotite. Reflecting these facts, the Ben Giang-Que Son granitic rocks were classified as syn-collision granite in our tectonic setting classification diagram (Figure 8) [54]. Some of the Ben Giang-Que Son granitic rocks could also be classified as adakitic rocks (Figure 9). Adakite is thought to be formed by the subduction of young oceanic crusts or ridges under relatively high temperatures [55]. The combined data suggested that the magma of the Ben Giang-Que Son granitic rocks was formed by the subduction of the Song Ma Ocean beneath the Indochina Block, then the magma separated and ascended through the mantle to reach the continental crust (Figure 13). The initial Sr isotope ratio values indicated that high proportions of continental crustal materials were incorporated at this time. Adakitic magma tends to produce I-type and magnetite-series igneous rocks [3,19,59]; however, the subsequent incorporation of Al-rich continental crustal materials and the reduction by carbon or graphite housed within these materials, such as clastic sedimentary rocks, produces I- to S-type and ilmenite-series granitic rocks. Most of the granitic rocks in the Loei Fold Belt, formed by the subduction of the Paleo-Tethys Ocean between the Sibumasu Block and the Indochina Block beneath the Indochina Block, are classified as adakitic rocks [11,19,59].

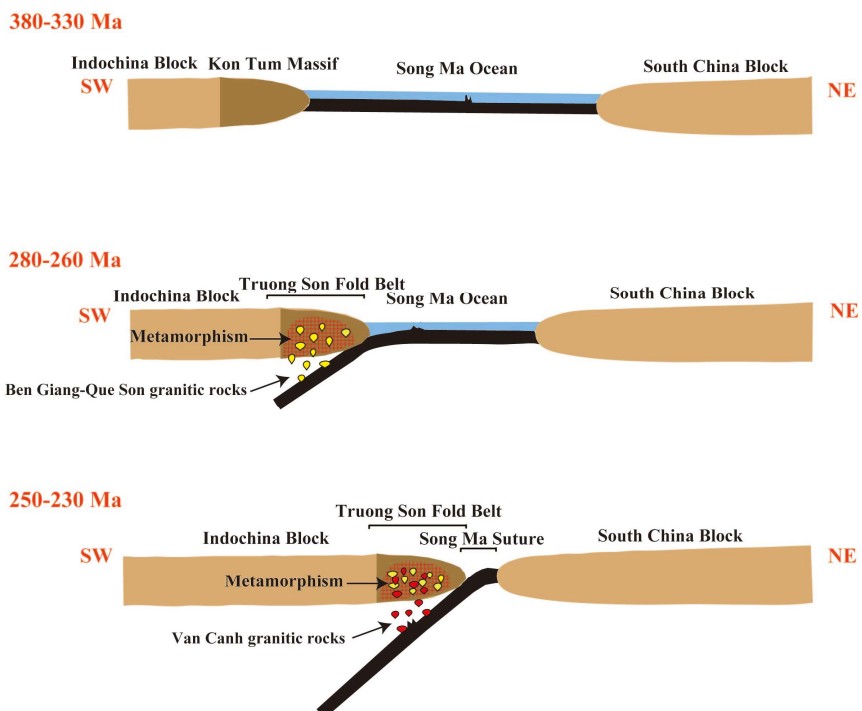

**Figure 13.** Schematic diagram of the tectonic evolution of the South China and Indochina blocks and the formation of the Van Canh (red) and Ben Giang-Que Son (yellow) granitic rocks.

On the other hand, the Van Canh granitic rocks were classified as I-type and magnetite-series granitic rocks. Hung et al. [28] considered the Van Canh granitic rocks to be S-type granite; however, this was an error. The incorporation of continental crustal materials was evident from their high initial Sr isotope ratios, but the amounts incorporated were relatively low compared to those in the Ben Giang-Que Son granitic rocks (Figure 12). This

was also reflected in the low A/CNK ratios and low total Al contents in the biotite in the Van Canh granitic rocks (Figure 11). Some of the Van Canh granitic rocks, as with the Ben Giang-Que Son granitic rocks, were formed from adakitic magma. However, the quantities of continental crustal materials incorporated into the Van Canh granitic rocks were smaller than those in the Ben Giang-Que Son granitic rocks. Adakitic magmatism has been reported in the Truong Son Fold Belt in Laos [3]. In addition, the Ben Giang-Que Son granitic rocks (ca. 280–260 Ma) [29], which intruded before the Van Canh granitic rocks (ca. 251–229 Ma) [28], caused high-temperature metamorphism within the continental crust (Figure 13). Hence, carbon or graphite in the surrounding continental crustal materials is thought to have been decomposed, to some extent, owing to the intrusion of the Ben Giang-Que Son granitic rocks. The later-intruding Van Canh granitic rocks, which incorporated continental crustal materials with less amount of carbon or graphite, were less reducing than the Ben Giang-Que Son granitic rocks. This was reflected by their ability to maintain the magnetic susceptibility of magnetite-series granitic rocks (Figure 12) [48]. These differences in petrogenesis and tectonic history ultimately led to the differences in geochemical signatures and magnetic susceptibility between the Ben Giang-Que Son and Van Canh granitic rocks.

## 6. Conclusions

Ben Giang-Que Son and Van Canh granitic rocks are widely distributed across the southern Kontum Massif. Their zircon U–Pb ages indicate that the Ben Giang-Que Son granitic rocks were formed during the Permian period (280–260 Ma) and that the Van Canh granitic rocks were formed during the Triassic period (251–229 Ma). Both the Ben Giang-Que Son and Van Canh granitic rocks were derived from magma of oceanic crust origin, generated by the subduction of the Song Ma Ocean, which is part of the Paleo-Tethys Ocean, beneath the Indochina Block. The Ben Giang-Que Son granitic magma was originally metaluminous but subsequently incorporated a large quantity of carbon or graphite-rich continental crustal materials, which chemically transformed the Ben Giang-Que Son granitic rocks from I-type and magnetite-series granitic rocks into I- to S-type and ilmenite-series granitic rocks. The Ben Giang-Que Son granitic rocks have high A/CNK ratios and high total Al contents in their biotite due to the incorporation of continental crustal materials. The intrusion of the Ben Giang-Que Son granitic rocks into the continental crust resulted in high-temperature metamorphic alterations, which decomposed some of the carbon or graphite contained in the surrounding continental crustal materials. The later-formed Van Canh granitic rocks were also derived from adakitic magma and incorporated continental crustal materials, but in smaller amounts compared to those in the Ben Giang-Que Son granitic rocks. As a result, the Van Canh granitic rocks have relatively low A/CNK ratios and low total Al contents in their biotite. In addition, much of the carbon or graphite in the continental crustal materials was already decomposed by the high-temperature metamorphism associated with the intrusion of the Ben Giang-Que Son granitic rocks; hence, the Van Canh granitic rocks retained I-type and magnetite-series signatures.

**Author Contributions:** Conceptualization, E.U.; methodology, E.U.; formal analysis, E.U., K.Y., T.Y. and N.M.; investigation, E.U., K.Y., T.Y. and N.M.; resources, E.U., K.Y., T.Y. and N.M.; data curation, E.U., K.Y., T.Y. and N.M.; writing—original draft preparation, E.U.; writing—reviewing and editing, E.U., K.Y., T.Y. and N.M.; visualization, E.U., K.Y., T.Y. and N.M.; supervision, E.U.; project administration, E.U.; funding acquisition, E.U. All authors have read and agreed to the published version of the manuscript.

**Funding:** This research was funded by a Waseda University Grant for Special Research Projects: 2022R-015.

**Data Availability Statement:** All data are included/referenced in this article.

**Acknowledgments:** This research was conducted with the support of a Joint Research Grant for Environmental Isotope Study of the Research Institute for Humanity and Nature. The authors are grateful to two anonymous reviewers for their insightful reviews and valuable comments to improve the quality of the manuscript.

**Conflicts of Interest:** The authors declare no conflict of interest.

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
