# Peer review of "Differences in Geochemical Signatures and Petrogenesis between the Van Canh and Ben Giang-Que Son Granitic Rocks in the Southern Kontum Massif, Vietnam"

_geosciences, doi:10.3390/geosciences13110341_

Round 1

Reviewer 1 Report

Comments and Suggestions for Authors

The authors presented whole-rock major and trace elements and Sr-Nd isotopes, and geochemical compositions of minerals for the granitic rocks from the Van Canh and Ben Giang-Que Son Granitic Rocks in the Southern Kon Tum Massif, Vietnam. The used these data to investigate the petrogenesis of the studied granitic rocks and their petrotectonic implications. Overall, the data reported in this study are of high quality, and are a useful supplement to the database of Permian-Triassic igneous rocks of Vietnam, the tectonic domain controlled by the Paleo-Tethys Ocean. However, there are some evident flaws in the paper (see comments below), publication of this paper in its present form in Geosciences is not recommended.

Major comments:

1.     All sections (abstract, introduction, results, discussions) need to be significantly improved, not only reorganization, but also the content.

For the abstract, some important result should be presented, the petrotectonic implications should be summarized and given.

For the introduction section, some parts belong to geological background, which is usually an individual part. The scientific question is not addressed. The logic needs to be considered and presented.

For the results part, this part needs to be reorganized, some sentences belong to discussion. And some data or values need to be described, such as K2O and P2O5.

For discussion section, 1) logic is very important. 2) Some references are not given, such as the ones supporting the organic matter and the peraluminous continental crust materials leading to the increasing Al in biotite. 3) Explanations on the organic matter and adakitic materials should be careful.

For conclusion, usually no references in this part. The important implications from your own study should be logically addressed here.

For figures, 1) the same color of the symbols; 2) the scale of the axis; 3) reference, such as the normalization value for fig. 10.

2. One of the highlight of your paper might be the classification of the granitic rocks developed in Van Canh and Ben Giang-Que Son. However, there are some evident flaws. 1) The classification scheme should be shown in the introduction part, as well as their meaning. 2) A/CNK is not the only or absolute classification indicator for I- and S-type granites. Some classical papers, such as Chappell et al. (2012 Lithos), should be cited. 3) the relationship between different classification scheme, especially the ones you mentioned in the paper, should be considered and explained.

3) The organic matter and adakitic materials should be explained in details.

Other comment:

1.     In fig. 4, Van Canh granitic rocks, an exception spot in the ilmenite series, why?

2.     In fig. 7 and 8, samples from both areas fall in the similar range.

3.     Although about four samples fall in the transitional field, only one sample in the adakite field in fig. 9. May be the reason why this one has high Sr/Y ratio should be considered, other than simply mention the adakitic material.

4.     In fig. 11, there are two groups of biotite in the Ben Giang-Que Son granitic rock.

5.     Fig. 12, one spot of Ben Giang-Que Son granitic rock has exceptionally high initial Sr ratios.

6.     Fig. 13, during this period, the Song Ma Ocean was subducted on the side of the South China Block or not?

Comments on the Quality of English Language

The English exposition needs significant improvement. Careful proofreading by a native English speaker/writer is needed before re-submission.

Author Response

Responses to the comments and suggestions from the Reviewer 1

Thank you for your helpful comments and suggestions to our manuscript submitted to Geosciences. We revised the manuscript taking them into consideration. Responses to your comments and suggestions are as follows;

Point 1: For the abstract, some important result should be presented, the petrotectonic implications should be summarized and given.

Response: The abstract has been rewritten to reflect the important results obtained in this study.

Point 2: For the introduction section, some parts belong to geological background, which is usually an individual part. The scientific question is not addressed. The logic needs to be considered and presented.

Response: The geological settings (background) were separated from “1. Introduction”. In addition to that, we have clarified the purpose of this study.

Point 3: For the results part, this part needs to be reorganized, some sentences belong to discussion. And some data or values need to be described, such as K2O and P2O5.

Response: We have tried to move as much as possible to "Discussion" what should be in "Discussion".

  A Na2O vs K2O diagram below was created to distinguish I-type granitic rocks from S-type granitic rocks, but unfortunately it did not yield useful results. We believe that an A/NK vs A/CNK diagram would be more effective for this purpose.

Although Chappell and White (1992) stated that I-type and S-type granitic rocks can be classified based on the content of P2O5 if the Rb content is greater than 250 ppm, the granitic rocks in this article all have Rb contents lower than 250 ppm and thus the content of P2O5 are not applicable for I-type and S-type classification.

Point 4: For discussion section, 1) logic is very important. 2) Some references are not given, such as the ones supporting the organic matter and the peraluminous continental crust materials leading to the increasing Al in biotite. 3) Explanations on the organic matter and adakitic materials should be careful.

Response: We revised “Discussion” taking into account what was pointed out, paying particular attention to the organic matter and adakite.

Point 5: For conclusion, usually no references in this part. The important implications from your own study should be logically addressed here.

Response: In the "Conclusions" section, we have tried to summarize the findings of this study without citing other articles.

Point 6: For figures, 1) the same color of the symbols; 2) the scale of the axis; 3) reference, such as the normalization value for fig. 10.

Response: The color of symbols and the scale of axis of figures have been unified. The reference related with the normalization value for Fig. 10 was described in the manuscript: McDonough and Sun [56].

Point 7:  One of the highlights of your paper might be the classification of the granitic rocks developed in Van Canh and Ben Giang-Que Son. However, there are some evident flaws. 1) The classification scheme should be shown in the introduction part, as well as their meaning. 2) A/CNK is not the only or absolute classification indicator for I- and S-type granites. Some classical papers, such as Chappell et al. (2012 Lithos), should be cited. 3) the relationship between different classification scheme, especially the ones you mentioned in the paper, should be considered and explained.

Response: In ”Introduction”, we showed the classification scheme for the two granitic rocks.

  As mentioned above (the response to the point 3), we believe that A/CNK is the most appropriate classification for I-type and S-type granitic rocks. The Na2O vs. K2O diagram is not suitable for the classification of I- and S-type. Chappell et al. (2012) was cited in the article.

    The relationship between different classification scheme was considered and explained in “Discussion”.

Point 8: The organic matter and adakitic materials should be explained in details.

Response: Metasedimentary rocks are abundant in Kontum Massif, and the source rocks, sedimentary rocks, are rich in organic matter. This was mentionned in the revised manuscript.

“adakitic materials” was replaced with “adakitic magmas”.

Point 9: In fig. 4, Van Canh granitic rocks, an exception spot in the ilmenite series, why?

Response: The sample VN307 shows similar magnetic susceptibility to magnetite-series granite on average, although weathering on the ground surface may have resulted in lower magnetic susceptibility in some locations.

Point 10: In fig. 7 and 8, samples from both areas fall in the similar range.

Response: The Van Canh granitic rocks plotted in a more typical “volcanic arc granite” area than the Ben Giang-Que Son granitic rocks in both figures..

Point 11: Although about four samples fall in the transitional field, only one sample in the adakite field in fig. 9. May be the reason why this one has high Sr/Y ratio should be considered, other than simply mention the adakitic material.

Response: The Van Canh and Ben Giang-Que Son granitic rocks are considered to be adakitic in origin. However, their Sr/Y ratios were reduced by incorporation of continental crustal materials. The extent of the Sr/Y ratio is thought to be governed by the amount of continental crustal materials incorporated. This was mentioned in the manuscript.

Point 12: In fig. 11, there are two groups of biotite in the Ben Giang-Que Son granitic rock.

Response: The total Al content in biotite may indicate that the Ben Giang-Que Son granitic rocks had a higher incorporation of continental crustal materials than the Van Canh granitic rocks.

Point 13: Fig. 12, one spot of Ben Giang-Que Son granitic rock has exceptionally high initial Sr ratios.

Response: The high initial Sr value may simply suggests that a large amount of continental crustal materials was involved in the magma formation.

Point 14: Fig. 13, during this period, the Song Ma Ocean was subducted on the side of the South China Block or not?

Response: It is supposed that the Song Ma Ocean did not subducted beneath the South China Block because there are no granitic rocks formed during the subduction of the Song Ma Ocean in the north side of the Song Ma Suture.

Reviewer 2 Report

Comments and Suggestions for Authors

The manuscript of Etsuo Uchida et al. provides detailed petrography, magnetic susceptibility, biotite chemical composition, and whole-rock geochemical and Nd‒Sr isotope composition data of the Van Canh and Ben Giang-Que Son granitic rocks from the Southern Kon Tum Massif in Vietnam. The authors have used the data to constrain the petrogenesis and tectonic history of both granitic rocks and clarify their differences in geochemical signatures and petrogenesis. The manuscript is well written and the topic is worth of publishing in Geosciences, but it raises concern at a very critical point for the entire story: the geological significance of organic matter in the formation of granitic rocks.

Granite can be formed by partial melting of old continental crust, on a local scale by in situ replacement of continental crust (granitization), by fractional crystallization of basalt magma, or by a combination of these processes. To date, organic matter (microbes) can be responsible for the weathering of granite and mechanisms involved at surface as well as it can promote REE mobilization, but microbe-granite interaction widely occurring on the Earth's surface remains imperfectly investigated (He et al., 2023).

He, YL, Ma, LY, Li, XR, Wang H, Liang XL, Zhu JX, He HP, 2023. Mobilization and fractionation of rare earth elements during experimental bio-weathering of granites. Geochimica et Cosmochimica Acta, Vol. 343, 384-395. https://doi.org/10.1016/j.gca.2022.12.027

I suggest that the authors improved some aspects of the manuscript based on the following comments:

1. Why do the authors insist in the text on the organic matter that they have not investigated and from which they are trying to constrain the petrogenesis and tectonic history of two suites of granites and clarify the differences in geochemical signatures and petrogenesis amongst both rocks? They would have simply analyzed the organic matter from these granites (if any???) and avoid any speculation, or also made a much more detailed summary of literary review on organic matter in granites!!! It therefore seems illusory and without solid foundations to suggest the contribution of organic matter for the petrogenesis and tectonic history of granites. The wide use of "organic matter" (9 times) which has not been surveyed is disconcerting!!!

2. In the Abstract section, the authors have mentioned that the organic matter was involved during the genesis of both suites of granites, but they have not presented evidence in regards with their results in the text.

2. In general, Introduction is essentially based on relevant background information related to the main topic of the study. In this case, the geological settings stand for introducing the significance of the study, which is rather confusing.

3. The section of geological setting is missing and should be clarified enough to acquaint the foreigner readers with the Southern Kon Tum Massif in Vietnam.

4. From Figure 1, Kon Tum (or Kontum???) seems to be a suture zone rather than a massif. How do you distinguish a massif in a tectonic map? Where is located the Tam Ky-Phuoc Son Suture on Figure 1? Please mark it in Figure 1.

5. It is not clear whether the studied granite samples are weathered or unweathered rocks. Please clarify it in the section of Materials and Methods.

6. Where was conducted in situ chemical analysis of biotite? Please clarify it in the text.

7. It is obvious that both set of granites could not be distinguished from macroscopic observation. Please delete the sentence in Line 135-136.

8. Except one Van Can granite sample, there is no obvious difference between Eu anomalies of Ben Giang-Que granites and those of Van Can granites (see Figure 10). The sentence is self-contradictory in Lines 195-199. Please rewrite it.

9. Negative Eu anomalies in granitic rocks indicate the removal of plagioclase and potassium feldspar from the fractionating magma rather than incorporation of organic matter in the continental crust materials involved in the genesis of granites.

10. Geochronological data of the studied granites are contradictory in the text.

Ben Giang-Que Son granites are thought to be of ca. 280‒260 Ma in line 49 and 278, while Van Canh granites are supposed to be of ca. 280‒206 Ma in line 293. On the other hand, Van Canh granites are thought to be of ca. 251‒229 Ma in lines 49 and 279, while Ben Giang-Que Son granites are supposed to be of ca. 251‒229 Ma in line 291-292. Please clarify it in the text.

Author Response

Responses to the comments and suggestions from the Reviewer 2

Thank you for your helpful comments and suggestions to our manuscript submitted to Geosciences. We revised the manuscript taking them into consideration. Responses to your comments and suggestions are as follows;

Point 1: Why do the authors insist in the text on the organic matter that they have not investigated and from which they are trying to constrain the petrogenesis and tectonic history of two suites of granites and clarify the differences in geochemical signatures and petrogenesis amongst both rocks? They would have simply analyzed the organic matter from these granites (if any???) and avoid any speculation, or also made a much more detailed summary of literary review on organic matter in granites!!! It therefore seems illusory and without solid foundations to suggest the contribution of organic matter for the petrogenesis and tectonic history of granites. The wide use of "organic matter" (9 times) which has not been surveyed is disconcerting!!!

Response: Granitic rocks are classified into I-type and S-type granites by Chappell and White (1974), as well as magnetite-series and ilmenite-series granitic rocks by Ishihara (1977). S-type granites are derived from sedimentary rocks, and because they contain organic matter (carbon) (Chapell and White, 1992), S-type granitic rocks form in reducing environments and must be ilmenite-series granitic rocks. Therefore, even granitic rocks of igneous rock origin (I-type granites) can change from oxidizing conditions (magnetite-series granites) to reducing conditions (ilmenite-series granites) by incorporation of sedimentary rocks.

  Of course, I-type and magnetite-series granitic rocks can be altered to S-type and ilmenite-series granitic rocks, respectively, by alteration of magnetite and other elements in the granitic rocks on the ground surface, but this is not essential.

Point 2: In the Abstract section, the authors have mentioned that the organic matter was involved during the genesis of both suites of granites, but they have not presented evidence in regards with their results in the text.

Response: Jiang et al. (2022) demonstrated that the Kontum Massif basement consists mainly of different units of metasedimentary rocks derived from sedimentary rocks deposited in five periods. We mentioned this in the Abstract section.

Point 3: In general, Introduction is essentially based on relevant background information related to the main topic of the study. In this case, the geological settings stand for introducing the significance of the study, which is rather confusing.

Response: The geological settings were separated from “1. Introduction”.

Point 4: The section of geological setting is missing and should be clarified enough to acquaint the foreigner readers with the Southern Kon Tum Massif in Vietnam.

Response: A new chapter "2. Geological Settings" has been created.

Point 5: From Figure 1, Kon Tum (or Kontum???) seems to be a suture zone rather than a massif. How do you distinguish a massif in a tectonic map? Where is located the Tam Ky-Phuoc Son Suture on Figure 1? Please mark it in Figure 1.

Response: Kon Tum has been unified with Kontum.

The Kontum Massif is not considered a Suture Zone, but the largest Precambrian basement exposure in the Indochina Block.

  The Tam Ky-Phuoc Son Suture was depicted in Fig. 1.

Point 6: It is not clear whether the studied granite samples are weathered or unweathered rocks. Please clarify it in the section of Materials and Methods.

Response: The following sentence was added:

Granitic rock samples were collected at roadside outcrops or quarries, selecting granitic rocks that were as fresh and as little weathered as possible during the field survey.

Point 7: Where was conducted in situ chemical analysis of biotite? Please clarify it in the text.

Response: As described in “2. Materials and Methods”, the chemical analysis of biotite was conducted using an energy dispersive spectrometer attached to a scanning microscope using rock thin sections.

Point 8: It is obvious that both set of granites could not be distinguished from macroscopic observation. Please delete the sentence in Line 135-136.

Response: We deleted the sentence in lines 135-136.

Point 9: Except one Van Can granite sample, there is no obvious difference between Eu anomalies of Ben Giang-Que granites and those of Van Can granites (see Figure 10). The sentence is self-contradictory in Lines 195-199. Please rewrite it.

Response: We consider that the samples VN307 and VN313 from the Van Canh granitic rocks show a small negative Eu anomaly, and the sample VN311 shows a positive Eu anomaly. This was described in the manuscript.

Point 10: Negative Eu anomalies in granitic rocks indicate the removal of plagioclase and potassium feldspar from the fractionating magma rather than incorporation of organic matter in the continental crust materials involved in the genesis of granites.

Response: The following sentences were added;

In magmas that have become reductive due to incorporation of organic matters, Eu3+ is reduced to Eu2+ and incorporated into plagioclase. As plagioclase is removed from the magma by differentiation, Eu3+ in the magma becomes relatively low, resulting in a negative Eu anomaly in the REE pattern.

Point 11: Geochronological data of the studied granites are contradictory in the text.

Ben Giang-Que Son granites are thought to be of ca. 280‒260 Ma in line 49 and 278, while Van Canh granites are supposed to be of ca. 280‒206 Ma in line 293. On the other hand, Van Canh granites are thought to be of ca. 251‒229 Ma in lines 49 and 279, while Ben Giang-Que Son granites are supposed to be of ca. 251‒229 Ma in lines 291-292. Please clarify it in the text.

Response: The correct formation age for the Van Canh granites is ca. 251-229 Ma, and that for the Ben Giang-Que Son granites is ca. 280-260 Ma.

Round 2

Reviewer 1 Report

Comments and Suggestions for Authors

1.     For the abstract, some important result should be presented, the petrotectonic implications should be summarized and given.

2.     For the introduction section, the scientific question is not addressed.

3.     Geological setting references should be given, such as  Line 75-78.

4.     Line 259, the normalization reference has been shown in the figure, do not need to write it in the text again.

5.     In the discussion part, why use adakitic magma to interpret the origin of I-type granites? Related references should be presented in the paper.

6.     Most of granites are derived from continental crust. The authors stress that the differences between studied granites are due to the proportion of continental crust materials. Does that mean those granites derived from mantle?

Comments on the Quality of English Language

English needs to be improved.

Author Response

Responses to the comments and suggestions from Reviewer 1

Thank you for your helpful comments and suggestions to our manuscript submitted to Geosciences. We revised our manuscript taking them into consideration as follows:

Point 1.     For the abstract, some important result should be presented, the petrotectonic implications should be summarized and given.

Response: We have revised “Abstract” to summarized the important results and petrotectonic implication.

Point 2.     For the introduction section, the scientific question is not addressed.

Response: We have rewritten the last sentence of "Intruduction" as follows:

On the basis of these data, we aimed to clarify the petrogenesis of both suites of granitic rocks in relation to the tectonic history between the South China Block, the Indochina Block, and the Song Ma Ocean.

Point 3.     Geological setting references should be given, such as Line 75-78.

Response: Geological setting references such as [4,5,13,15,22] were input in Line 75-78.

Point 4.     Line 259, the normalization reference has been shown in the figure, do not need to write it in the text again.

Response: As is the case with the other figures, the references are written in the text and also in the figures, so Line 259 remained as it is.

Point 5.     In the discussion part, why use adakitic magma to interpret the origin of I-type granites? Related references should be presented in the paper.

Response: Adakitic magma, produced by subduction of high-temperature oceanic crust, produces magnetite-series and I-type igneous rocks. Not all I-type granitic rocks are formed by adakitic magmas. We cited references [3,19,59] as an example of how adakitic granitic rocks produced by subduction of high-temperature oceanic crust are of magnetite-series and I-type.

Point 6.     Most of granites are derived from continental crust. The authors stress that the differences between studied granites are due to the proportion of continental crust materials. Does that mean those granites derived from mantle?

Response: The back-arc side of the oceanic crust subduction zone becomes an extensional field, making it easier for mantle-derived magma to rise, and in such locations, magnetite-series and I-type granitic rocks are formed. The Loei Fold Belt, generated by the subduction of the Paleo-Tethys Ocean beneath the Indochina Block, is a typical example of this [19,59]. On the fore-arc side (for example Sukhothai Zone), I- to S-type and ilmenite-series granitic rocks are formed from continental crustal material [19,59].

Reviewer 2 Report

Comments and Suggestions for Authors

The revised manuscript of Etsuo Uchida et al. has been improved. However, there are still fundamental problems with the solid scientific foundations for the contribution of organic matter in the petrogenesis of the Ben Giang-Que Son granites. The authors do not report compelling petrological, geochemical, and isotopic evidence for the source of organic matter incorporated in the metasedimentary rocks from which these granitic rocks were formed. Thus, the contribution of organic matter for the petrogenesis of the studied granites is still questionable.

In response to previous point 2 requesting evidence of organic matter mentioned by the authors in their Abstract, they argued that “Jiang et al. (2022) demonstrated that the Kontum Massif basement consists mainly of different units of metasedimentary rocks derived from sedimentary rocks deposited in five periods.” However, Jiang et al. (2022) didn’t report the presence of organic matter in metasedimentary rocks of the Kontum Massif basement, neither incorporation of organic matter in sediments.

Amongst the questions satisfactorily answered for the S-type granitic magmas was the nature of their source rocks as follows (see Clemens, 2003): “The most appropriate source materials of S-type magmas would be metagreywackes, including labile volcaniclastic types and some metatonalites (biotite –plagioclase–quartz rocks).”

Clemens, J.D., 2003. S-type granitic magmas—petrogenetic issues, models and evidence. Earth-Science Reviews 61, 1–18.

Carbon refers to non-metallic element rather than organic matter.

Author Response

Responses to the comments and suggestions from Reviewer 2

Thank you for your helpful comments and suggestions to our manuscript submitted to Geosciences. We revised our manuscript taking them into consideration as follows:

Point 1: The revised manuscript of Etsuo Uchida et al. has been improved. However, there are still fundamental problems with the solid scientific foundations for the contribution of organic matter in the petrogenesis of the Ben Giang-Que Son granites. The authors do not report compelling petrological, geochemical, and isotopic evidence for the source of organic matter incorporated in the metasedimentary rocks from which these granitic rocks were formed. Thus, the contribution of organic matter for the petrogenesis of the studied granites is still questionable.

Response: Ishihara (1981) [48] and Chapell and White (1992) [49] noted that S-type and ilmenite-series granitic rocks formed under relatively reducing conditions are attributable to organic matter in sedimentary rocks.

Point 2: In response to previous point 2 requesting evidence of organic matter mentioned by the authors in their Abstract, they argued that “Jiang et al. (2022) demonstrated that the Kontum Massif basement consists mainly of different units of metasedimentary rocks derived from sedimentary rocks deposited in five periods.” However, Jiang et al. (2022) didn’t report the presence of organic matter in metasedimentary rocks of the Kontum Massif basement, neither incorporation of organic matter in sediments.

Response: The presence of organic matter in sedimentary rocks, which are the source rocks of metasedimentary rocks, is a common phenomenon. However, the lack of organic matter in metasedimentary rocks formed from sedimentary rocks may be the result of metamorphism.

Point 3: Amongst the questions satisfactorily answered for the S-type granitic magmas was the nature of their source rocks as follows (see Clemens, 2003): “The most appropriate source materials of S-type magmas would be metagreywackes, including labile volcaniclastic types and some metatonalites (biotite –plagioclase–quartz rocks).”

Clemens, J.D., 2003. S-type granitic magmas—petrogenetic issues, models and evidence. Earth-Science Reviews 61, 1–18.

Response: Our paper does not address the formation of typical S-type granitic rocks. However, we are certain that the source material of S-type granitic rocks is sedimentary rocks in continental crustal materials.

Point 4: Carbon refers to non-metallic element rather than organic matter.

Response: We have rewritten it to “organic matter and/or carbon”.

Round 3

Reviewer 1 Report

Comments and Suggestions for Authors

No comment.

Author Response

Responses to the comments and suggestions from Reviewer 1

Thank you for your insightful review and valuable comments to enhance the quality of our submitted manuscript.

Reviewer 2 Report

Comments and Suggestions for Authors

While it is true that there are organically formed sedimentary rocks, organic matter is not the source of sedimentary rocks. Note that there are three main types of sedimentary rocks known as clastic, chemical, and bioclastic rocks. Conclusively, the presence of organic matter is not a common phenomenon in sedimentary rocks.

The interpretations of the authors are not consistent with the interpretation of Chappell and White (1992), who suggested that "the generally more reduced character of the S-type granites is thought to be due to the presence of graphite in the sedimentary source rocks, ..., but it could be due to the intrisic nature of the structures of metaluminous and peraluminous melts (e.g. Mysen & Virgo 1989)."

I recommend the authors changing "organic matter" to "graphite" in the whole manuscript unless they provide compelling evidence of organic matter involved in the formation of the studied granites.

Comments on the Quality of English Language

No comment

Author Response

Responses to the comments and suggestions from Reviewer 2

Thank you for your comments and suggestions to our manuscript submitted to “geosciences”. We revised our manuscript taking them into consideration as follows:

Point 1: While it is true that there are organically formed sedimentary rocks, organic matter is not the source of sedimentary rocks. Note that there are three main types of sedimentary rocks known as clastic, chemical, and bioclastic rocks. Conclusively, the presence of organic matter is not a common phenomenon in sedimentary rocks.

Response: “sedimentary rocks” was replaced by “clastic sedimentary rocks”.

Point 2: The interpretations of the authors are not consistent with the interpretation of Chappell and White (1992), who suggested that "the generally more reduced character of the S-type granites is thought to be due to the presence of graphite in the sedimentary source rocks, ..., but it could be due to the intrisic nature of the structures of metaluminous and peraluminous melts (e.g. Mysen & Virgo 1989)."

I recommend the authors changing "organic matter" to "graphite" in the whole manuscript unless they provide compelling evidence of organic matter involved in the formation of the studied granites.

Response: “organic matter” was replaced by “carbon or graphite”.